# VERLET FLOWS: EXACT-LIKELIHOOD INTEGRATORS FOR FLOW-BASED GENERATIVE MODELS

**Ezra Erives, Bowen Jing, Tommi Jaakkola**
CSAIL, Massachusetts Institute of Technology
{erives,bjing}@mit.edu, tommi@csail.mit.edu

## ABSTRACT

Approximations in computing model likelihoods with continuous normalizing flows (CNFs) hinder the use of these models for importance sampling of Boltzmann distributions, where exact likelihoods are required. In this work, we present *Verlet flows*, a class of CNFs on an augmented state-space inspired by symplectic integrators from Hamiltonian dynamics. When used with carefully constructed *Taylor-Verlet integrators*, Verlet flows provide exact-likelihood generative models which generalize coupled flow architectures from a non-continuous setting while imposing minimal expressivity constraints. On experiments over toy densities, we demonstrate that the variance of the commonly used Hutchinson trace estimator is unsuitable for importance sampling, whereas Verlet flows perform comparably to full autograd trace computations while being significantly faster.

## 1 INTRODUCTION

Flow-based generative models—also called *normalizing flows*—parameterize maps from prior to data distributions via invertible transformations. An exciting application of normalizing flows is in learning the Boltzmann distributions of physical systems (Noé et al., 2019; Midgley et al., 2023; Kim et al., 2024). At inference time, these *Boltzmann generators* provide model likelihoods which can be used to reweigh samples towards the target energy with importance sampling. While nearly all existing Boltzmann generators are built from composing invertible layers such as coupling layers or splines, experiments on image domains suggest that *continuous* normalizing flows (CNFs)—which can parameterize arbitrary vector fields mapping noise to data—are far more expressive than their discrete counterparts (Chen et al., 2018; Grathwohl et al., 2018). Unfortunately, the exact model likelihood of CNFs can only be accessed through expensive trace computations and numerical integration, preventing their adoption in Boltzmann generators.

In this work, we propose *Verlet flows*, a flexible class of CNFs on an augmented state-space inspired by symplectic integrators from Hamiltonian dynamics. Instead of parameterizing the flow $\gamma$ with a single neural network, Verlet flows instead parameterize the coefficients of the multivariate Taylor expansions of $\gamma$ in both the state-space and the augmenting space. We then introduce *Taylor-Verlet integrators*, which exploit the splitting approximation from which many symplectic integrators are derived to approximate the intractable time evolution of $\gamma$ as the composition of the tractable time evolutions of the Taylor expansion terms. At training time, Verlet flows are a subclass of CNFs, and can be trained accordingly. At inference time, *Taylor-Verlet integration* enables theoretically-sound importance sampling with exact likelihoods.

## 2 BACKGROUND

**Discrete Normalizing Flows** Given a source distribution $\pi_0$ and target distribution $\pi_1$, we wish to learn an invertible, bijective transformation $f_\theta$ which maps $\pi_0$ to $\pi_1$. Discrete normalizing flows parameterize $f_\theta$ as the composition $f_\theta = f_\theta^N \circ \cdots \circ f_\theta^i$, from which $\log \pi_1(f_\theta(x))$ can be computed using the change of variables formula and the log-determinants of the Jacobians of the individual transformations $f_\theta^i$. Thus, significant effort has been dedicated to developing expressive, invertible building blocks $f_\theta^i$ whose Jacobians have tractable log-determinant. Successful approaches include *coupling-based* flows, in which the dimensions of the state variable $x$ are partitioned in two, and the

each half is used in turn to update the other half (Dinh et al., 2016; 2014; Müller et al., 2019; Durkan et al., 2019), and *autoregressive* flows (Kingma et al., 2017; Papamakarios et al., 2018). Despite these efforts, discrete normalizing flows have been shown to suffer from a lack of expressivity in practice.

**Continuous Normalizing Flows** Continuous normalizing flows (CNFs) dispense with the discrete layers of normalizing flows and instead learn a time-dependent vector field $\gamma(x, t; \theta)$, parameterized by a neural network, which maps the source $\pi_0$ to a target distribution $\pi_1$ (Chen et al., 2018; Grathwohl et al., 2018). Model densities can be accessed by the *continuous-time* change of variables formula given by

$$\log \pi_1(x_1) = \log \pi_0(x_0) - \int_0^1 \mathrm{Tr}\, J_\gamma(x_t, t; \theta)\, dt, \tag{1}$$

where $x_t = x_0 + \int_0^t \gamma(x_t, t; \theta)\, dt$, $\mathrm{Tr}$ denotes trace, and $J_\gamma(x_t, t; \theta) = \frac{\partial \gamma(x, t; \theta)}{\partial x}\big|_{x_t, t}$ denotes the Jacobian. Compared to discrete normalizing flows, CNFs are not constrained by invertibility or the need for a tractable Jacobian, and therefore enjoy significantly greater expressivity.

While the trace $\mathrm{Tr}\, J_\gamma(x_t, t; \theta)$ appearing in the integrand of Equation 1 can be evaluated exactly with automatic differentiation, this grows prohibitively expensive as the dimensionality of the data grows large, as a linear number of backward-passes are required. In practice, the Hutchinson trace estimator (Grathwohl et al., 2018) is used to provide a linear-time, unbiased estimator of the trace. While cheaper, the variance of the Hutchinson estimator makes it unsuitable for importance sampling.

**Symplectic Integrators and the Splitting Approximation** Leap-frog integration is a numeric method for integrating Newton's equations of motion which involves alternatively updating $q$ (position) and $p$ (velocity) in an invertible manner not unlike augmented, coupled normalizing flows.[1] Leap-frog integration is a special case of the more general family of *symplectic integrators*, designed for the Hamiltonian flow $\gamma_H$ (of which the equations of motion are a special case). Oftentimes the Hamiltonian flow decomposes as $\gamma_H = \gamma_q + \gamma_p$, enabling the *splitting approximation*

$$\varphi(\gamma_H, \tau) \approx \varphi(\gamma_q, \tau) \circ \varphi(\gamma_p, \tau) \tag{2}$$

where $\varphi(\gamma, \tau)$ denotes the time evolution operator along the flow $\gamma$ for a duration $\tau$, and where the terms on the right-hand side of Equation 2 are possibly tractable in a way that the left-hand side is not. For example, the leap-frog integrator corresponds to analytic, invertible, and volume-preserving $\varphi(\gamma_{\{q,p\}}, t)$, whereas the original evolution may satisfy none of these properties. While Verlet flows, to be introduced in the next section, are not in general Hamiltonian, they similarly exploit the splitting approximation. A more detailed exposition of symplectic integrators and the splitting approximation can be found in Appendix A.

## 3 METHODS

### 3.1 VERLET FLOWS

We consider the problem of mapping a source distribution $\tilde{\pi}_0(q)$ on $\mathbb{R}^{d_q}$ at time $t = 0$ to a target distribution $\tilde{\pi}_1(q)$ on $(\mathbb{R}^{d_q})$ at time $t = 1$ by means of a time-dependent flow $\gamma(x, t)$. We will now augment this problem on the configuration-space $\mathbb{R}^{d_q}$ by extending the distribution $\tilde{\pi}_0(q)$ to $\pi_0(q, p) = \pi_0(p|q)\tilde{\pi}_0(q)$ and $\tilde{\pi}_1(q)$ to $\pi_1(q, p) = \pi_1(p|q)\tilde{\pi}_1(q)$ where both $\pi_i(p|q)$ are given by $\mathcal{N}(p; 0, I_{d_p})$. In analogy with Hamiltonian dynamics, we will refer to the space $M = \mathbb{R}^{d_q + d_p}$ as *phase space*.[2]

Observe that any analytic flow $\gamma$ is given (at least locally) by a multivariate Taylor expansion of the form

$$\gamma(x, t) = \frac{d}{dt} \begin{bmatrix} q \\ p \end{bmatrix} = \begin{bmatrix} \gamma^q(q, p, t) \\ \gamma^p(q, p, t) \end{bmatrix} = \begin{bmatrix} s_0^q(p, t) + s_1^q(p, t)^T q + \cdots \\ s_0^p(q, t) + s_1^p(q, t)^T p + \cdots \end{bmatrix} = \begin{bmatrix} \sum_{k=0}^\infty s_k^q(p, t)(q^{\otimes k}) \\ \sum_{k=0}^\infty s_k^p(q, t)(p^{\otimes k}) \end{bmatrix} \tag{3}$$

for appropriate choices of functions $s_i^q$ and $s_i^p$, which we have identified in the last equality as $(i, 1)$-tensors: multilinear maps which take in $i$ copies of $q \in T_q \mathbb{R}^n$ and return a tangent vector. While

---

[1] Closely related to leap-frog integration is *Verlet integration*, from which our method derives its name.
[2] Note that we do not require that $d_q = d_p$.

$s_0^{\{q,p\}}$ and $s_1^{\{q,p\}}$ can be thought of as vectors and matrices respectively, higher order terms do not admit particularly intuitive interpretations. Whereas traditional CNFs commonly parameterize $\gamma_\theta$ directly via a neural network, *Verlet flows* instead parameterize the coefficients $s_k^{\{q,p\};\theta}$ with neural networks, allowing for Verlet integration via the splitting approximation. By parameterizing all the terms in the Taylor expansion, Verlet flows are in theory as expressive as CNFs parameterized as $\gamma(q, p, t; \theta)$. However, in practice, we must truncate the series after some finite number of terms, yielding the order $N$ Verlet flow

$$\gamma_N(x, t; \theta) \coloneqq \begin{bmatrix} \sum_{k=0}^{N} s_k^q(p, t; \theta)(q^{\otimes k}) \\ \sum_{k=0}^{N} s_k^p(q, t; \theta)(p^{\otimes k}) \end{bmatrix}. \tag{4}$$

In the next section, we examine how to obtain exact likelihoods from these truncated Verlet flows.

### 3.2 TAYLOR-VERLET INTEGRATORS

Denote by $\gamma_k^q$ the flow given by

$$\gamma_k^q(x, t; \theta) = \begin{bmatrix} s_k^q(p, t; \theta)(q^{\otimes k}) \\ 0 \end{bmatrix} \in T_x M,$$

and define $\gamma_k^p$ similarly.[3] For any such flow $\gamma'$ on $M$, denote by $\varphi^\ddagger(\gamma', \tau)$ the *time evolution operator*, transporting a point $x \in M$ along the flow $\gamma'$ for time $\tau$. We denote by just $\varphi$ the *pseudo* time evolution operator given by $\varphi(\gamma', \tau) : x_t \to x_t + \int_t^{t+\tau} \gamma'(x_s, t)\, ds$.[4] Note that $t$ is kept constant throughout integration, an intentional choice which we shall see allows for a tractable closed form. Although our Verlet flows are not Hamiltonian, the splitting approximation from Equation 11 can be applied to Verlet flows to decompose the desired time evolution into simpler, analytic terms, yielding

$$\varphi^\ddagger(\gamma, \tau) \approx \varphi(\gamma_t, \tau) \circ \varphi(\gamma_N^p, \tau) \circ \varphi(\gamma_N^q, \tau) \circ \varphi(\gamma_{N-1}^p, \tau) \circ \varphi(\gamma_{N-1}^q, \tau) \cdots \varphi(\gamma_0^p, \tau) \circ \varphi(\gamma_0^q, \tau). \tag{5}$$

Note here that the leftmost term of the right hand side is the time-update term $\varphi(\gamma_t, \tau)$. The key idea is that Equation 5 **approximates the generally intractable $\varphi^\ddagger(\gamma, \tau)$ as a composition of simpler, tractable updates allowing for a closed-form, exact-likelihood integrator for Verlet flows**.

The splitting approximation from Equation 5, together with closed-form expressions for the time evolution operators and their log density updates (see Figure 1), yields an integration scheme specifically tailored for Verlet flows, and which we shall refer to as a *Taylor-Verlet integrator*. Explicit integrators for first order and higher order Verlet flows are presented in Appendix D. One important element of the design space of Taylor-Verlet integration is the order of the terms within the splitting approximation of Equation 5, and consequently, the order of updates performed during Verlet integration. We will refer to Taylor-Verlet integrators which follow the order of Equation 5 as *standard* Taylor-Verlet integrators, and others as non-standard. While the remainder of this work focuses on standard Taylor-Verlet integrators, the space of non-standard Taylor-Verlet integrators is rich and requires further exploration. Certain coupling-based normalizing flow architectures, such as RealNVP (Dinh et al., 2016) can be realized as the update steps of non-standard Taylor-Verlet integrators, as is discussed in Appendix E.

### 3.3 CLOSED FORM AND DENSITY UPDATES FOR TIME EVOLUTION OPERATORS

For each pseudo time evolution operator $\varphi(\gamma_{\{q,p\}}^k, \tau)$, we compute its closed-form and the log-determinant of its Jacobian. Together, these allow us to implement the integrator given by Equation 5. Results are summarized in the Table 1 for $\gamma_k^q$ only, but analogous results hold for for $\gamma_k^p$ as well. Note that for terms of order $k \geq 2$, and for the sake of tractability, we restrict our attention to *sparse* tensors, denoted $\overline{s_k}^{\{q,p\}}$, for which only "on-diagonal" terms are non-zero so that $\overline{s_k}^{\{q,p\}}(q^{\otimes k})$ collapses to a simple dot product. We similarly use $\overline{\gamma}_k^{\{q,p\}}$ to denote the corresponding flows for sparse, higher order terms. Full details and derivations can be found in Appendix C.

---

[3] When there is no risk of ambiguity, we drop the subscript and refer to $\gamma_N$ simply by $\gamma$.

[4] Justification for use of the pseudo time evolution operator $\varphi$ can be found in Appendix B.

Table 1: A summary of closed-forms for the time evolution operators $\varphi(\gamma_k^q; \tau)$, and their corresponding log density updates. Analogous results hold for for $\varphi(\gamma_k^p; \tau)$ as well.

| Flow $\gamma$ | Operator $\varphi(\gamma, \tau)$ | Density Update $\log \det |J\varphi(\gamma, \tau)|$ |
|---|---|---|
| $\gamma_0^q$ | $\begin{bmatrix} q \\ p \end{bmatrix} \to \begin{bmatrix} q + \tau s_0^q(p, t) \\ p \end{bmatrix}$ | $0$ |
| $\gamma_1^q$ | $\begin{bmatrix} q \\ p \end{bmatrix} \to \begin{bmatrix} \exp(\tau s_1^q(p, t))q \\ p \end{bmatrix}$ | $\mathrm{Tr}(\tau s_1^q(p, t))$ |
| $\overline{\gamma}_k^q, k > 1$ | $\begin{bmatrix} q \\ p \end{bmatrix} \to \begin{bmatrix} (q^{\circ(1-k)} + \tau(\overline{s}_k^q)_i(1-k))^{\circ\left(\frac{1}{1-k}\right)} \\ p \end{bmatrix}$ | $\sum_i \frac{k}{1-k} \log\left|q_i^{1-k} + \tau(1-k)(\overline{s}_k^q)_i\right| - k \log|q_i|$ |

## 4 EXPERIMENTS

Across all experiments in this section, and unless stated otherwise, we train an order-one Verlet flow $\gamma_\theta$, with coefficients $s_{0,1}^{\{q,p\};\theta}$ parameterized as a three-layer architecture with $64$ hidden units each, as a continuous normalizing flow using likelihood-based loss. Non-Verlet integration is performed numerically using a fourth-order Runge-Kutta solver for $100$ steps.

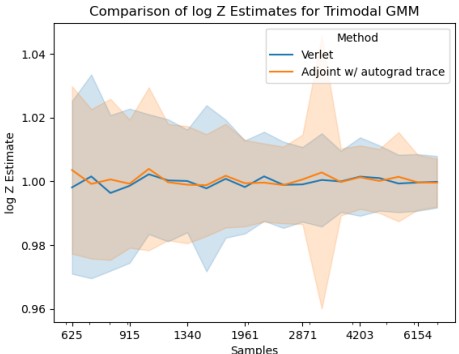
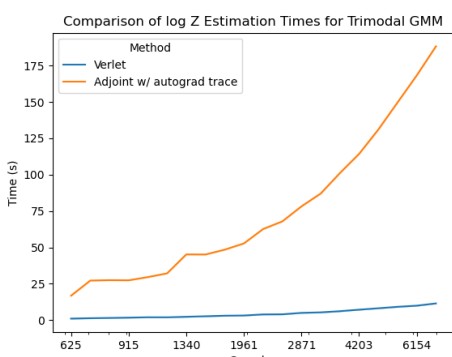

Figure 1: The left graph shows estimates of the natural logarithm $\log Z$ (mean $\pm$ S.D.) as a function of the number of samples. The right graph shown the time needed to make the computations in the left graph. Both graphs use $100$ integration steps.

**Estimation of** $\log Z$    Given an unnormalized density $\widehat{\pi}$, a common application of importance sampling is to estimate the partition function $Z = \int \widehat{\pi}(x)\, dx$. Given a distribution $\pi_\theta$ (hopefully close to the unknown, normalized density $\pi = \frac{\widehat{\pi}}{Z}$), we obtain an unbiased estimate of $Z$ via

$$\mathbb{E}_{x \sim \pi_\theta}\left[\frac{\widehat{\pi}(x)}{\pi_\theta(x)}\right] = \int_{\mathbb{R}^d}\left[\frac{\widehat{\pi}(x)}{\pi_\theta(x)}\right]\pi_\theta(x)\, dx = \int_{\mathbb{R}^d} \widehat{\pi}(x)\, dx = Z. \tag{6}$$

We train an order-one Verlet flow $\gamma_\theta$ targeting a trimodal Gaussian mixture in two-dimensional $q$-space, and an isotropic Gaussian $\mathcal{N}(p_1; 0, I_2)$ in a two-dimensional $p$-space. We then perform and time importance sampling using Equation 6 to estimate the natural logarithm $\log Z$ in two ways: first numerically integrating $\gamma_\theta$ with a fourth-order Runge-Kutta solver and using automatic differentiation to exactly compute the trace, and secondly using Taylor-Verlet integration. We find that integrating $\gamma_\theta$ using a Taylor-Verlet integrator performs comparably to integrating numerically while being significantly faster. Results are summarized in Figure 1.

The poor performance of the Hutchinson trace estimator can be seen in Figure 2, where we plot a histogram of the logarithm $\log\left[\frac{\widehat{\pi}(x)}{\pi_\theta(x)}\right]$ of the importance weights for $x \sim \pi_\theta(x)$. The presence of just a few positive outliers (to be expected given the variance of the trace estimator) skews the resulting estimate of $Z$ to be on the order of $10^{20}$ or larger.

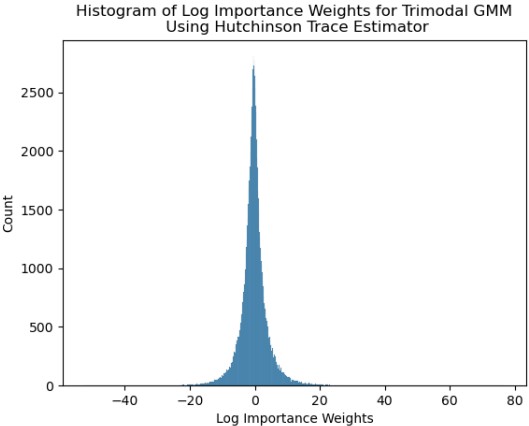

Figure 2: This histogram shows log importance weights for a trimodal GMM obtained by numerically integrating the Verlet flow $\gamma_\theta$ using the Hutchinson trace estimator for 100 integration steps. Positive outliers render the Hutchinson trace estimator unusable for importance sampling.

## 5 CONCLUSION

In this work, we have presented Verlet flows, a class of CNFs in an augmented state space whose flow $\gamma_\theta$ is parameterized via the coefficients of a multivariate Taylor expansion. The splitting approximation used by many symplectic integrators is adapted to construct exact-likelihood Taylor-Verlet integrators, which enable comparable but faster performance to numeric integration using expensive, autograd-based trace computation on tasks such as importance sampling.

## 6 ACKNOWLEDGEMENTS

We thank Gabriele Corso, Xiang Fu, Peter Holderrieth, Hannes Stärk, and Andrew Campbell for helpful feedback and discussion over the course of the project. We also thank the anonymous reviewers for their helpful feedback and suggestions.

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

## A    HAMILTONIAN MECHANICS AND SYMPLECTIC INTEGRATORS ON EUCLIDEAN SPACE

Given a mechanical system with configuration space $\mathbb{R}^d$, we may define the *phase space* of the system to be the cotangent bundle $M = T^*\mathbb{R}^d \simeq \mathbb{R}^{2d}$. Intuitively, phase space captures the intuitive notion that understanding the state of $M$ at a point in time requires knowledge of both the position $q \in \mathbb{R}^d$ and the velocity, or momentum (assuming unit mass), $p \in T^*\mathbb{R}^d$.

### A.1    HAMILTONIAN MECHANICS

*Hamiltonian mechanics* is a formulation of classical mechanics in which the equations of motion are given by differential equations describing the flow along level curves of an energy function, or *Hamiltonian*, $\mathcal{H}(q, p)$. Denote by $\mathcal{X}(M)$ the space of smooth vector fields on $M$. Then at the point $(q, p) \in M$, the *Hamiltonian flow* $\gamma_\mathcal{H} \in \mathcal{X}(M)$ is defined to be the unique vector field which satisfies

$$\gamma_\mathcal{H}^T \Omega \gamma' = \nabla \mathcal{H} \cdot \gamma' \tag{7}$$

for all $\gamma' \in \mathcal{X}(M)$, and where

$$\Omega = \begin{bmatrix} 0 & I_d \\ -I_d & 0 \end{bmatrix}$$

is the *symplectic form*[5]. Equation 7 implies $\gamma_\mathcal{H}^T \Omega = \nabla \mathcal{H}$, which yields

$$\gamma_\mathcal{H} = \begin{bmatrix} \frac{\partial \mathcal{H}}{\partial p} & -\frac{\partial \mathcal{H}}{\partial q} \end{bmatrix}^T. \tag{8}$$

In other words, our state $(q, p)$ evolves according to $\frac{dq}{dt} = \frac{\partial \mathcal{H}}{\partial p}$ and $\frac{dp}{dt} = -\frac{\partial \mathcal{H}}{\partial q}$.

### A.2    PROPERTIES OF THE HAMILTONIAN FLOW $\gamma_\mathcal{H}$

The time evolution $\varphi^\ddagger(\gamma_\mathcal{H}, \tau)$ of $\gamma_\mathcal{H}$ satisfies two important properties: it conserves the Hamiltonian $\mathcal{H}$, and it conserves the symplectic form $\Omega$.

**Proposition A.1.** *The flow $\gamma_\mathcal{H}$ conserves the Hamiltonian $\mathcal{H}$.*

*Proof.* This amounts to showing that $\frac{d}{d\tau} \varphi^\ddagger(\gamma_\mathcal{H}, \tau)|_{\tau=0} = 0$, which follows immediately from $\nabla \mathcal{H} \cdot \gamma_\mathcal{H} = 0$. ☐

**Proposition A.2.** *The flow $\gamma_\mathcal{H}$ preserves the symplectic form $\Omega$.*

---

[5]In our Euclidean context, a symplectic form is more generally any non-degenerate skew-symmetric bilinear form $\Omega'$ on phase space. However, it can be shown that there always exists a change of basis which satisfies $\Lambda \Omega' \Lambda^{-1} = \Omega$, where $\Lambda$ denotes the change of basis matrix. Thus, we will only consider $\Omega$.

*Proof.* Realizing $\Omega$ as the (equivalent) two-form $\sum_i dq_i \wedge dp_i$, the desired result amounts to showing that the Lie derivative $\mathcal{L}_{\gamma_\mathcal{H}} \Omega = 0$. With Cartan's formula, we find that

$$\mathcal{L}_{\gamma_\mathcal{H}} \Omega = d(\iota_{\gamma_\mathcal{H}} \Omega) + \iota_{\gamma_\mathcal{H}} d\Omega = d(\iota_{\gamma_\mathcal{H}} \Omega)$$

where $d$ denotes the exterior derivative, and $\iota$ denotes the interior product. Here, we have used that $d\Omega = \sum_i d(dq_i \wedge dp_i) = 0$. Then we compute that

$$d(\iota_{\gamma_\mathcal{H}} \Omega) = d(\iota_{\gamma_\mathcal{H}} \sum_i dq_i \wedge dp_i)$$
$$= d\left( \sum_i \frac{\partial \mathcal{H}}{\partial p_i} dp_i + \frac{\partial \mathcal{H}}{\partial q_i} dq_i \right)$$
$$= d(d\mathcal{H}).$$

Since $d^2 = 0$, $\mathcal{L}_{\gamma_\mathcal{H}} = d(d\mathcal{H}) = 0$, as desired. $\qquad \square$

Flows which preserve the symplectic form $\Omega$ are known as *symplectomorphisms*. Proposition A.2 implies that the time evolution of $\gamma_H$ is a symplectomorphism.

### A.3 SYMPLECTIC INTEGRATORS AND THE SPLITTING APPROXIMATION

We have seen that the time-evolution of $\gamma_\mathcal{H}$ is a symplectomorphism, and therefore preserves the symplectic structure on the phase space $M$. In constructing numeric integrators for $\gamma_\mathcal{H}$, it is therefore desirable that our integrators are, if possible, themselves symplectomorphisms. In many cases, the Hamiltonian $\mathcal{H}$ decomposes as the sum $\mathcal{H}(q, p) = T(q) + V(p)$. Then, at the point $z = (q, p) \in M$, we find that

$$\gamma_T = \begin{bmatrix} \frac{\partial T}{\partial p} \\ -\frac{\partial T}{\partial q} \end{bmatrix} = \begin{bmatrix} 0 \\ -\frac{\partial T}{\partial q} \end{bmatrix} \in T_z(\mathbb{R}^2)$$

and

$$\gamma_V = \begin{bmatrix} \frac{\partial V}{\partial p} \\ -\frac{\partial V}{\partial q} \end{bmatrix} = \begin{bmatrix} \frac{\partial V}{\partial p} \\ 0 \end{bmatrix} \in T_z(\mathbb{R}^2).$$

Thus, the flow decomposes as well to

$$\gamma_\mathcal{H} = \begin{bmatrix} \frac{\partial \mathcal{H}}{\partial p} \\ -\frac{\partial \mathcal{H}}{\partial q} \end{bmatrix} = \begin{bmatrix} \frac{\partial V}{\partial p} \\ -\frac{\partial T}{\partial q} \end{bmatrix} = \begin{bmatrix} 0 \\ -\frac{\partial T}{\partial q} \end{bmatrix} + \begin{bmatrix} \frac{\partial \mathcal{H}}{\partial p} \\ 0 \end{bmatrix} = \gamma_T + \gamma_V.$$

Observe now that the respective time evolution operators are tractable and are given by

$$\varphi^\ddagger(\gamma_T, \tau) : \begin{bmatrix} q \\ p \end{bmatrix} \to \begin{bmatrix} q + \tau \frac{\partial T}{\partial p} \\ p \end{bmatrix}$$

and

$$\varphi^\ddagger(\gamma_V, \tau) : \begin{bmatrix} q \\ p \end{bmatrix} \to \begin{bmatrix} q \\ p - \tau \frac{\partial T}{\partial q} \end{bmatrix}.$$

Since $\gamma_T$ and $\gamma_V$ are Hamiltonian flows their time evolutions $\varphi^\ddagger(\gamma_T, \tau)$ and $\varphi^\ddagger(\gamma_T, \tau)$ are both symplectomorphisms. As symplectomorphisms are closed under composition, it follows that that $\varphi^\ddagger(\gamma_T, \tau) \circ \varphi^\ddagger(\gamma_V, \tau)$ is itself a symplectomorphism. We have thus arrived at the *splitting approximation*

$$\varphi^\ddagger(\gamma_\mathcal{H}, \tau) \approx \varphi^\ddagger(\gamma_T, \tau) \circ \varphi^\ddagger(\gamma_V, \tau). \tag{9}$$

Equation 9 allows us to approximate the generally intractable, symplectic time evolution $\varphi^\ddagger(\gamma_\mathcal{H}, \tau)$ as the symplectic composition of two simpler, tractable time evolution operators. The integration scheme given by Equation 9 is generally known as the *symplectic Euler method*.

So-called splitting methods make use of more general versions of the splitting approximation to derive higher order, symplectic integrators. Using the same decomposition $\mathcal{H}(q, p) = T(q) + V(p)$, and instead of considering the two-term approximation given by Equation 9, we may choose

coefficients $\{c_i\}_{i=0}^N$ and $\{d_i\}_{i=0}^N$ with $\sum c_i = \sum d_i = 1$ and consider the more general splitting approximation

$$\varphi^{\ddagger}(\gamma_{\mathcal{H}}, \tau) \approx \varphi^{\ddagger}(c_N \gamma_T) \circ \varphi^{\ddagger}(d_N \gamma_V) \circ \cdots \circ \varphi^{\ddagger}(c_0 \gamma_T) \circ \varphi^{\ddagger}(d_0 \gamma_V). \tag{10}$$

A more detailed exposition of higher order symplectic integrators can be found in (Yoshida, 1993).

## B  JUSTIFICATION FOR TREATING $\varphi(\gamma, \tau)$'S AS TIME EVOLUTION OPERATORS

In the following discussion, we will use $x_t = (q_t, p_t)$ for brevity. The splitting approximation from Equation 5, which we recall below as

$$\varphi^{\ddagger}(\gamma, \tau) \approx \varphi(\gamma_t, \tau) \circ \varphi(\gamma_N^p, \tau) \circ \varphi(\gamma_N^q, \tau) \cdots \varphi(\gamma_0^p, \tau) \circ \varphi(\gamma_0^q, \tau). \tag{11}$$

requires some clarification. Recall that while the *true* time evolution operator $\varphi^{\ddagger}(\gamma, \tau)$ is given by

$$\varphi^{\ddagger}(\gamma, \tau) : \begin{bmatrix} x_t \\ t \end{bmatrix} \rightarrow \begin{bmatrix} x_t + \int_t^{t+\tau} \gamma(x_u, u)\, du \\ t + \tau \end{bmatrix}, \tag{12}$$

the pseudo time operator $\varphi(\gamma, \tau)$ is given by

$$\varphi(\gamma, \tau) : \begin{bmatrix} x_t \\ t \end{bmatrix} \rightarrow \begin{bmatrix} x_t + \int_t^{t+\tau} \gamma(x_u, t)\, du \\ t \end{bmatrix}, \tag{13}$$

where $t$ is kept-constant throughout the integration.

To make sense of the connection between $\varphi^{\ddagger}$ and $\varphi$, we will augment our phase-time space $\mathcal{S} = \mathbb{R}^{d_p + d_q} \times \mathbb{R}_{\geq 0}$ (within which our points $(x_t, t)$ live), with a new $s$-dimension, to obtain the space $\mathcal{S}' = \mathcal{S} \times \mathbb{R}_{\geq 0}$. Treating $x_t$ and $t$ as the state variables $x_s$ and $t_s$ which evolve with $s$, the flow $\gamma_k^q$ (as a representative example) on $\mathbb{R}^{d_p + d_q}$ can be extended to a flow $\widehat{\gamma}_k^q$ on $\mathcal{S}$ given by

$$\widehat{\gamma}_k^q(x_s, t_s) = \begin{bmatrix} \frac{\partial x_s}{\partial s} \\ \frac{\partial t_s}{\partial s} \end{bmatrix} = \begin{bmatrix} \gamma_k^q(x_s, t_s) \\ 0 \end{bmatrix} \tag{14}$$

where the zero $t_s$-component encodes the fact that the pseudo-time evolution $\varphi(\gamma_k^q, \tau)$ from Equation 13 does not change $t$. The big idea is then that this pseudo time evolution $\varphi(\gamma_k^q, \tau)$ can be viewed as the projection of the (non-pseudo) $s$-evolution $\varphi^{\ddagger}(\widehat{\gamma}_k^q, \tau)$, given by

$$\varphi^{\ddagger}(\widehat{\gamma}_k^q, \tau) : \begin{bmatrix} x_s \\ t_s \\ s \end{bmatrix} \rightarrow \begin{bmatrix} x_s + \int_s^{s+\tau} \gamma_k^q(x_u, t_u)\, du \\ t_{s+\tau} \\ s + \tau \end{bmatrix}, \tag{15}$$

onto $\mathcal{S}$. The equivalency follows from the fact that for $\widehat{\gamma}_k^q$, $t_{s+\tau'} = t_s$ for $\tau' \in [0, \tau]$. A similar statement can be made about the $t$-update $\gamma_t$ from Equation 11.

Denoting by $\mathrm{Proj} : \mathcal{S}' \rightarrow \mathcal{S}$ the projection onto $\mathcal{S}$, we see that the splitting approximating using pseudo-time operators from Equation 11 can be rewritten as the projection onto $S$ of an analogous splitting approximation using non-pseudo $s$-evolution operators, viz.,

$$\mathrm{Proj}\, \varphi^{\ddagger}(\widehat{\gamma}, \tau) \approx \mathrm{Proj}\, \left[ \varphi^{\ddagger}(\widehat{\gamma}_t, \tau) \circ \varphi^{\ddagger}(\widehat{\gamma}_N^p, \tau) \circ \varphi^{\ddagger}(\widehat{\gamma}_N^q, \tau) \cdots \varphi^{\ddagger}(\widehat{\gamma}_0^p, \tau) \circ \varphi^{\ddagger}(\widehat{\gamma}_0^q, \tau) \right]. \tag{16}$$

## C  DERIVATION OF TIME EVOLUTION OPERATORS AND THEIR JACOBIANS

**Order Zero Terms.**  For order $k = 0$, recall that

$$\gamma_0^q(x) = \begin{bmatrix} s_0^q(p, t)(q^{\otimes 0}) \\ 0 \end{bmatrix} = \begin{bmatrix} s_0^q(p, t) \\ 0 \end{bmatrix},$$

so that the operator $\varphi(\gamma_q^0, \tau)$ is given by

$$\varphi(\gamma_0^q, \tau) : \begin{bmatrix} q \\ p \\ t \end{bmatrix} \rightarrow \begin{bmatrix} q + \tau s_0^q(p,t) \\ p \\ t \end{bmatrix} \tag{17}$$

with Jacobian $J_0^q$ given by

$$J_0^q = \begin{bmatrix} I_{d_q} & \tau(\frac{\partial s_0^q}{\partial p})^T & \tau(\frac{\partial s_0^q}{\partial t})^T \\ 0 & I_{d_p} & 0 \\ 0 & 0 & 1 \end{bmatrix}. \tag{18}$$

The analysis for $s_0^p$ is nearly identical, and we omit it.

**Order One Terms.** For $k = 1$, we recall that

$$\gamma_1^q(x) = \begin{bmatrix} s_1^q(p,t)(q^{\otimes 1}) \\ 0 \\ 0 \end{bmatrix} = \begin{bmatrix} s_1^q(p,t)^T q \\ 0 \\ 0 \end{bmatrix}. \tag{19}$$

Then the time evolution operator $\varphi(\gamma_1^q, \tau)$ is given by

$$\varphi(\gamma_1^q, \tau) : \begin{bmatrix} q \\ p \\ t \end{bmatrix} \rightarrow \begin{bmatrix} \exp(\tau s_1^q(p,t))q \\ p \\ t \end{bmatrix} \tag{20}$$

and the Jacobian $J_1^q$ is simply given by

$$J_1^q = \begin{bmatrix} \exp(\tau s_1^q(p,t)) & \cdots & \cdots \\ 0 & I_{d_p} & 0 \\ 0 & 0 & 1 \end{bmatrix} \tag{21}$$

Then $\log\det(J_q^1) = \log\det(\exp(\tau a_1(p,t))) = \log\exp(\mathrm{Tr}(\tau a_1(p,t))) = \mathrm{Tr}(\tau a_1(p,t))$.

**Sparse Higher Order Terms.** For $k > 1$, we consider only sparse tensors given by the simple dot product

$$\bar{s}_k^q(q^{\otimes k}) = \sum_i (\bar{s}_k^q)_i \, q_i^k = \left(\bar{s}_k^q(q^{\otimes k})\right)^T q^{\circ k}$$

where $q^{\circ k}$ denotes the element-wise $k$-th power of $q$. Then the $q$-component of time evolution operator $\overline{\gamma}_k^q$ is given component-wise by an ODE of the form $\frac{dq}{dt} = s_k^q(p,t)q^k$, whose solution is obtained in closed form via rearranging to the equivalent form

$$\int_{q_t}^{q_{t+\tau}} \frac{1}{\bar{s}_k^q(p,t)} q^{-k} \, dq = \int_t^{t+\tau} dt = \tau.$$

Then it follows that $q_{t+\tau}$ is given component-wise by $(q_{t,i}^{1-k} + \tau \bar{s}_k^q(p,t)_i(1-k))^{\frac{1}{1-k}}$. Thus, the operator $\varphi(\overline{\gamma}_k^q, \tau)$ is given by

$$\varphi(\overline{\gamma}_k^q, \tau) : \begin{bmatrix} q \\ p \\ t \end{bmatrix} \rightarrow \begin{bmatrix} \left(q^{\circ(1-k)} + \tau \bar{s}_k^q(p,t)(1-k)\right)^{\circ\left(\frac{1}{1-k}\right)} \\ p \\ t \end{bmatrix}. \tag{22}$$

The Jacobian is then given by

$$J_k^q = \begin{bmatrix} \mathrm{diag}\left(q^{-k}\left(q^{\circ(1-k)} + \tau\bar{s}_k^q(p,t)(1-k)\right)^{\circ\left(\frac{1}{1-k}-1\right)}\right) & \cdots & \cdots \\ 0 & I_{d_p} & 0 \\ 0 & 0 & 1 \end{bmatrix} \tag{23}$$

with $\log\det|J_k^q|$ given by

$$\log\det\mathrm{diag}\left|q^{\circ-k}\left(q^{\circ(1-k)} + \tau\bar{s}_k^q(p,t)(1-k)\right)^{\circ\left(\frac{k}{1-k}\right)}\right| = \sum_i \frac{k}{1-k}\log|q_i^{1-k} - \tau s_k^q(p,t)_i(1-k)| - k\log|q_i|.$$

## D    EXPLICIT DESCRIPTIONS OF TAYLOR-VERLET INTEGRATORS

Taylor-Verlet integrators are constructed using the splitting approximation given in Equation 5 of an order $N$ Verlet flow $\gamma_\theta$, which we recall below as

$$\varphi^\ddagger(\gamma, \tau) \approx \varphi(\gamma_t, \tau) \circ \varphi(\gamma_N^p, \tau) \circ \varphi(\gamma_N^q, \tau) \cdots \varphi(\gamma_0^p, \tau) \circ \varphi(\gamma_0^q, \tau). \tag{24}$$

The standard Taylor-Verlet integrator of an order $N$ Verlet flow $\gamma_\theta$ is given explicitly in Algorithm 1 below.

---
**Algorithm 1** Integration of order $N$ Verlet flow
---
1: **procedure** ORDERNVERLETINTEGRATE($q, p, t_0, t_1, \text{steps}, \gamma_\theta, N$)
2:     $\tau \leftarrow \frac{t_1 - t_0}{\text{steps}}, t \leftarrow t_0$
3:     $\Delta \log p = 0$                                                   ▷ Change in log density.
4:     $s_0^q, s_0^p, \ldots s_N^q, s_N^p \leftarrow \gamma_\theta$
5:     **while** $t < t_1$ **do**
6:         $k \leftarrow 0$
7:         **while** $k \leq N$ **do**
8:             $q \leftarrow \varphi(\gamma_k^{q;\theta}, \tau)$                          ▷ $q$-update.
9:             $\Delta \log p \leftarrow \Delta \log p - \log \det J\varphi(\gamma_k^{q;\theta}, \tau)$
10:             $p \leftarrow \varphi(\gamma_k^{p;\theta}, \tau)$                          ▷ $p$-update.
11:             $\Delta \log p \leftarrow \Delta \log p - \log \det J\varphi(\gamma_k^{p;\theta}, \tau)$
12:             $k \leftarrow k + 1$
13:         $t \leftarrow t + \tau$
14:     **return** $q, p, \Delta \log p$
---

Closed-form expressions for the time evolution operators $\gamma_k^{q;\theta}, \tau)$ and log density updates $\log \det J\varphi(\gamma_k^{q;\theta}, \tau)$ can be found in Table 1. Algorithm 2 details explicitly standard Taylor-Verlet integration of an order one Verlet flow.

---
**Algorithm 2** Integration of order one Verlet flow
---
1: **procedure** ORDERONEVERLETINTEGRATE($q, p, t_0, t_1, \text{steps}, \gamma_\theta$)
2:     $\tau \leftarrow \frac{t_1 - t_0}{\text{steps}}, t \leftarrow t_0$
3:     $\Delta \log p = 0$                                                   ▷ Change in log density.
4:     $s_0^q, s_0^p, s_1^q, s_1^p \leftarrow \gamma_\theta$
5:     **while** $t < t_1$ **do**
6:         $q \leftarrow q + \tau s_0^q(p, t; \theta),$                          ▷ Apply equation 17
7:         $p \leftarrow p + \tau s_0^p(q, t; \theta)$                          ▷ Apply equation 17
8:         $q \leftarrow \exp(\tau s_1^q(p, t; \theta))q$                          ▷ Apply equation 20
9:         $\Delta \log p \leftarrow \Delta \log p - \text{Tr}(\tau s_1^q(p, t; \theta))$          ▷ Apply equation 23
10:         $p \leftarrow \exp(\tau s_1^p(q, t; \theta))p$                          ▷ Apply equation 20
11:         $\Delta \log p \leftarrow \Delta \log p - \text{Tr}(\tau s_1^p(q, t; \theta))$          ▷ Apply equation 23
12:         $t \leftarrow t + \tau$
13:     **return** $q, p, \Delta \log p$
---

## E    REALIZING COUPLING ARCHITECTURES AS VERLET INTEGRATORS

In this section, we will show that two coupling-based normalizing flow architectures - NICE (Dinh et al. (2014)) and RealNVP (Dinh et al. (2016)) - can be realized as the Taylor-Verlet integrators for zero and first order Verlet flows respectively. Specifically, for each such coupling layer architecture $f_\theta$, we may construct a Verlet flow $\gamma_\theta$ whose Taylor-Verlet integrator is given by successive applications of $f_\theta$.

**Additive Coupling Layers**  The additive coupling layers of NICE involve updates of the form

$$f_\theta^q(q, p) = \text{concat}(q + t_\theta^q(p), p),$$
$$f_\theta^p(q, p) = \text{concat}(q, p + t_\theta^p(q)).$$

Now consider the order zero Verlet flow $\gamma_\theta$ given by

$$y_\theta = \frac{1}{\tau} \begin{bmatrix} \tilde{t}_\theta^q(p, t) \\ \tilde{t}_\theta^p(q, t) \end{bmatrix},$$

where $\tilde{t}_\theta^q(x, t) \triangleq t_\theta^q(x)$ and $\tilde{t}_\theta^p(x, t) \triangleq t_\theta^p(x)$. Then the standard Taylor-Verlet integrator with step size $\tau$ is given by the splitting approximation

$$\varphi^\ddagger(\gamma_\theta, \tau) \approx \varphi(\gamma_t, \tau) \circ \varphi(\gamma_p^{0;\theta}, \tau) \circ \varphi(\gamma_q^{0;\theta}, \tau)$$

with updates given by

$$\varphi(\gamma_q^{0;\theta}, \tau) : \begin{bmatrix} q \\ p \end{bmatrix} \to \begin{bmatrix} q + (\tau)\left(\frac{1}{\tau}\tilde{t}_\theta^q(p, t)\right) \\ p \end{bmatrix} = \begin{bmatrix} q + t_\theta(p) \\ p \end{bmatrix}$$

and

$$\varphi(\gamma_p^{0;\theta}, \tau) : \begin{bmatrix} q \\ p \end{bmatrix} \to \begin{bmatrix} q \\ p + (\tau)\left(\frac{1}{\tau}\tilde{t}_\theta^p(q, t)\right) \end{bmatrix} = \begin{bmatrix} q \\ p + t_\theta(q) \end{bmatrix}.$$

Thus, $f_\theta^q = \varphi(\gamma_q^{0;\theta}, \tau)$ and $f_\theta^q = \varphi(\gamma_q^{0;\theta}, \tau)$.

**RealNVP**  The coupling layers of RealNVP are of the form

$$f_\theta^q(q, p) = \text{concat}(q \odot \exp(s_\theta^q(p)) + t_\theta^q(p), p),$$
$$f_\theta^p(q, p) = \text{concat}(q, p \odot \exp(s_\theta^p(q)) + t_\theta^p(q).$$

Now consider the first order Verlet flow $\gamma_\theta$ given by

$$\gamma_\theta = \begin{bmatrix} \tilde{t}_\theta^q + (\tilde{s}_\theta^q)^T q \\ \tilde{t}_\theta^p + (\tilde{s}_\theta^p)^T p \end{bmatrix},$$

where $\tilde{s}_\theta^q(p, t) := \frac{1}{\tau}\text{diag}(s_\theta^q(p))$,

$$\tilde{t}_\theta^q(p, t) := \frac{t_\theta^q(p)}{\tau \exp(\tau \tilde{s}_\theta^q(p))},$$

and $\tilde{s}_\theta^p$ and $\tilde{t}_\theta^p$ are defined analogously. Then a non-standard Taylor-Verlet integrator is obtained from the splitting approximation

$$\varphi^\ddagger(\gamma_\theta, \tau) \approx \varphi(\gamma_t, \tau) \circ \varphi(\gamma_p^{1;\theta}, \tau) \circ \varphi(\gamma_p^{0;\theta}, \tau) \circ \varphi(\gamma_q^{1;\theta}, \tau) \circ \varphi(\gamma_q^{0;\theta}, \tau)$$

where the order has been rearranged from that of Equation 5 to group together the $\gamma^q$ and $\gamma^p$ terms. The time evolution operators $\varphi(\gamma_q^{0;\theta}, \tau)$ and $\varphi(\gamma_q^{1;\theta}, \tau)$ are given by

$$\varphi(\gamma_q^{0;\theta}, \tau) : \begin{bmatrix} q \\ p \end{bmatrix} \to \begin{bmatrix} q + \tau \tilde{t}_\theta^q(p, t) \\ p \end{bmatrix} = \begin{bmatrix} q + \frac{t_\theta^q(p)}{\exp(\tau \tilde{s}_\theta^q(p, t))} \\ p \end{bmatrix}$$

and

$$\varphi(\gamma_q^{1;\theta}, \tau) : \begin{bmatrix} q \\ p \end{bmatrix} \to \begin{bmatrix} \exp(\tau \tilde{s}_\theta^q(p, t))^T q \\ p \end{bmatrix}.$$

So that the combined $q$-update $\varphi(\gamma_q^{1;\theta}, \tau) \circ \varphi(\gamma_q^{0;\theta}, \tau)$ is given by

$$\varphi(\gamma_q^{1;\theta}, \tau) \circ \varphi(\gamma_q^{0;\theta}, \tau) : \begin{bmatrix} q \\ p \end{bmatrix} \to \begin{bmatrix} \exp(\tau \tilde{s}_\theta^q(p, t))^T q + t_\theta^q(p) \\ p \end{bmatrix} = \begin{bmatrix} \exp(\text{diag}(s_\theta^q(p))^T q + t_\theta^q(p) \\ p \end{bmatrix}$$

which reduces to

$$\begin{bmatrix} q \odot \exp(s_\theta^q(p)) + t_\theta^q(p) \\ p \end{bmatrix} = \text{concat}(q \odot \exp(s_\theta^q(p)) + t_\theta^q(p), p) = f_\theta^q(q, p).$$

Thus, $f_\theta^q(q, p) = \varphi(\gamma_q^{1;\theta}, \tau) \circ \varphi(\gamma_q^{0;\theta}, \tau)$, and similarly, $f_\theta^p(q, p) = \varphi(\gamma_p^{1;\theta}, \tau) \circ \varphi(\gamma_p^{0;\theta}, \tau)$.

Strictly speaking, Taylor-Verlet integrators cannot be said to completely generalize these coupling-based architectures because Verlet flows operate on a fixed, canonical partition of dimensions, whereas coupling-based architectures commonly rely on different dimensional partitions in each layer.

