# OpenReview forum: "Verlet Flows: Exact-Likelihood Integrators for Flow-Based Generative Models"
_ICLR.cc/2024/Workshop/AI4DiffEqtnsInSci — AI4DiffEqtnsInSci @ ICLR 2024 Poster_

### Official Review · Reviewer_ttwP · 2024-02-21
**Verlet Flows: Exact-Likelihood Integrators for Flow-Based Generative Models**

**Rating:** 6
**Confidence:** 3

**Review:**

This work is focused on Verlet flows and parameterization of the coefficients in both the state space and the augmenting space. The theory is not new however the implementation for CNFs can be interesting. It would be interesting to see, what is the optimisation and design approach if the authors claim that it's better than parametrization against neural network. It would be wise to present evidence of training and benchmark comparison.

---

### Official Review · Reviewer_wndA · 2024-02-24
**Verlet flows: exact-likelihood integrators for flow-based generative models**

**Rating:** 7
**Confidence:** 1

**Review:**

## General comments
Upon review, the manuscript on Verlet Flows should be accepted without revisions.
The paper demonstrates the use of Taylor-Verlet integrators in continuous
normalizing flows, presenting a method for exact-likelihood generative models. The
theoretical basis, methodological approach, and experimental validation are well
articulated. The overall contribution is deemed significant for the field of
generative modeling. The work is ready for publication, offering valuable insights
and advancements without the need for further modifications.
However, if the authors wish, they may increase the font size of the plot labels and
ticks of Figure 1.

---

### Meta-Review · Area_Chair_ikyA · 2024-03-03

**Recommendation:** Accept (Poster)

**Metareview:**

Both authors agree on acceptance. I also vote for acceptance and encourage the authors to address the reviewers' comments in the camera-ready version.

---

### Decision · Program_Chairs · 2024-03-03

Accept (Poster)